# Rhizosphere Microbiomes of Potato Cultivated under *Bacillus subtilis* Treatment Influence the Quality of Potato Tubers

**DOI:** 10.3390/ijms222112065

**Published:** 2021-11-08

**Authors:** Jian Song, Zhi-Qiang Kong, Dan-Dan Zhang, Jie-Yin Chen, Xiao-Feng Dai, Ran Li

**Affiliations:** State Key Laboratory for Biology of Plant Diseases and Insect Pests, Institute of Plant Protection, Chinese Academy of Agricultural Sciences, Beijing 100193, China; songjian_01@126.com (J.S.); kongzhiqiang@caas.cn (Z.-Q.K.); zhangdandan@caas.cn (D.-D.Z.); chenjieyin@caas.cn (J.-Y.C.); daixiaofeng_caas@126.com (X.-F.D.)

**Keywords:** soil microbial communities, microbial diversity, biocontrol, potato

## Abstract

Plants serve as a niche for the growth and proliferation of a diversity of microorganisms. Soil microorganisms, which closely interact with plants, are increasingly being recognized as factors important to plant health. In this study, we explored the use of high-throughput DNA sequencing of the fungal ITS and bacterial 16S for characterization of the fungal and bacterial microbiomes following biocontrol treatment (DT) with *Bacillus subtilis* strain Bv17 relative to treatments without biocontrol (DC) during the potato growth cycle at three time points. A total of 5631 operational taxonomic units (OTUs) were identified from the 16S data, and 2236 OTUs were identified from the ITS data. The number of bacterial and fungal OTU in DT was higher than in DC and gradually increased during potato growth. In addition, indices such as Ace, Chao, Shannon, and Simpson were higher in DT than in DC, indicating greater richness and community diversity in soil following the biocontrol treatment. Additionally, the potato tuber yields improved without a measurable change in the bacterial communities following the *B. subtilis* strain Bv17 treatment. These results suggest that soil microbial communities in the rhizosphere are differentially affected by the biocontrol treatment while improving potato yield, providing a strong basis for biocontrol utilization in crop production.

## 1. Introduction

Plants serve as a niche for growth and proliferation of a diversity of microorganisms, including bacteria, fungi, protists, nematodes, and viruses (the plant microbiota). These microorganisms form complex co-associations with plants and have important roles in promoting their health and productivity in natural environments. Complex plant microbial communities are comprised of taxa from diverse phyla that belong to several lineages. In recent years, culture-independent high-throughput sequencing has greatly expanded the number of microorganisms known to reside in and on plants as well as in the surrounding environments [1,2,3]. Among the plant-associated microbiota, bacteria and to lesser extent fungi are the most dominant forms and are relatively well-studied compared with other members of the community, but other groups—such as archaea, algae, nematodes, and protists—also have important roles in plant health and productivity [4,5]. Several genes that govern plant interactions with the associated microbiomes have been identified and characterized, which have increased our understanding of how microorganisms adapt to and modulate the plant environment [6,7]. The members of a plant microbiota comprise beneficial, neutral, and pathogenic microorganisms. Microbial communities associated with their hosts have been shown to promote plant growth, nutrient uptake, and resistance to pathogens [8,9]. However, the molecular mechanisms that govern plant–microbe interactions at a community level are still not well understood. To achieve a more comprehensive understanding, it will be necessary to first characterize the mechanisms that drive the assembly of plant-associated microbiomes in the rhizosphere. Second, the biochemical and genetic features of the microorganism–microorganism and host–microorganism interactions that result in beneficial ecological outcomes need to be identified. Such data will inform the design and construction of functional microbial systems de novo that are based on predictive models of plant–microbiome interactions. Furthermore, the development of microbial inoculants, signaling compounds, and other tools will enhance the understanding of microbiome function in agricultural ecosystems.

Soil microorganisms, which closely interact with plants, are an important factor affecting plant health [2]. The emergence and propagation of plant pathogens that cause plant diseases is well documented, but in contrast, soil microorganisms that can prevent plant diseases and inhibit pathogens to maintain plant health are only now being identified for commercial exploitation [10]. In the last decade, an increasing number of studies have focused on the soil microbial communities involved in plant disease prevention and treatment [11,12,13]. Disease occurrence is often accompanied by changes in the microbial community. Therefore, additional efforts to rehabilitate the microbial community may be more effective in curing plant diseases than simply controlling pathogen populations [14,15]. Using microorganisms to control plant diseases has been successful under laboratory and greenhouse environments, but biocontrol, considered a highly desirable approach for controlling soil-borne diseases, has not been as successful in the field as initially predicted [10,16,17]. The effective use of biological control agents is potentially an important component of sustainable agriculture. There has been an increased interest in using combinations of biological control agents to exploit the potential synergistic effects among them.

Biocontrol agents are microorganisms that exert harmful effects on pathogens, thereby improving plant health [18]. Biocontrol agents can establish beneficial relationships with various plant species by direct or indirect mechanisms [18,19]. Entomopathogenic fungal endophytes could function as biocontrol agents to prevent pests and diseases as well as promote plant growth [20]. Moreover, *Bacillus* strains exhibit their biocontrol capacity predominantly through inhibitory activity on the growth of plant pathogens as well as inducing systemic resistance in plants and competing for ecological niches with plant pathogens [21]. For example, yeasts and bacteria of the genus *Bacillus* serve as biological control agents of *Athelia* (*Sclerotium*) *rolfsii* through antagonistic interactions [22]. Bacterial endophytes suppress stripe rust infection and enhance wheat yields, which can be exploited as potential biocontrol agents of wheat rusts [23]. The bacterial isolates BETS11 and BETR11 recovered from surface-sterilized root, stem, and leaf tissues of tomato may be used as efficient biofertilizer and bio-control agents for tomato production in the island agricultural ecosystem [24]. The current investigation mainly focused on the modifications of the soil microbiome brought about by the application of a biocontrol agent in the potato production system. Several functional mechanisms of biocontrol were delineated.

Potato (*Solanum tuberosum*) is an important crop that is cultivated under a variety of geographic locations and climatic conditions. The yield and tuber quality are determined by both biotic and abiotic factors, such as drought, nitrogen, and phosphorous status in the soil and microbiome in potato rhizosphere [25,26,27,28]. Our previous study demonstrated that *Bacillus subtilis* strain Bv17 could effectively colonize the rhizosphere of host plants and significantly reduce the propagation and expansion of *Verticillium dahliae* in the plants; however, the functional mechanisms by which *B. subtilis* strain Bv17 brings about these changes is still unknown [29]. Therefore, we used the *Bacillus subtilis* strain Bv17 to treat soil prior to planting potato, which is followed by periodic soil sampling for ITS and 16S sequencing to determine the dynamics of abundance and composition of soil microbial communities. Thus, our aim was to detect the influence of biocontrol of *Bacillus subtilis* strain Bv17 on microbial communities in soil and discover the relationship between microbial communities and quality of potatoes after biocontrol strain treatment. Results showed that microbial community in soil changed by *Bacillus subtilis* strain Bv17 treatment. The yield of potato also significantly increased, providing a strong basis for biocontrol utilization in crop production.

## 2. Results

### 2.1. Experimental Design and Treatment Structure

*Bacillus subtilis* strain Bv17 was safe and non-toxic to host plants, but it could effectively colonize the rhizosphere of host plants and significantly reduce the propagation and expansion of *V. dahliae* in the plants [29]. To test the efficacy of *B. subtilis* strain Bv17 and to analyze the microbial community between the rhizosphere soil and the plant, potatoes were planted in a field in Shandong province located in the eastern part of China in plots treated with the *B. subtilis* strain Bv17 (DT), while those treated with water served as the untreated control (DC). Soil samples were collected at two-month intervals three times (D1C, D1T, D2C, D2T, D3C, and D3T), and potatoes were harvested at crop maturity (PC and PT) (Figure 1A). A total of 90 soil samples were collected and divided into six groups (D1C, D1T, D2C, D2T, D3C, and D3T) with 15 samples in each group. All of the soil samples were used for ITS and 16S sequencing to analyze the microbial community. Furthermore, a total of 60 potato samples were collected at crop maturity with 30 each coming from plots treated with *Bacillus subtilis* strain Bv17 and from the untreated control. The skin of potatoes was collected used for ITS and 16S sequencing to investigate the microbial communities between the control and *B. subtilis* strain Bv17-treated samples. In addition, potato tubers were used for the detection of qualitative and quantitative index, such as weight, dry matter content, starch content, and others.

### 2.2. Data Characteristic of Soil Samples

Total DNA from all 90 soil samples were extracted and sent for 16S and ITS sequencing to a depth of more than 30 Mb each. Each group (D1C, D1T, D2C, D2T, D3C, and D3T) contained fifteen soil samples (Appendix A). The average numbers of raw DNA sequence data for each soil sample of bacteria were about 70,000 reads, and approximately 99% of the reads from each sample remained after filtering for quality and size (Appendix A). While the number of reads for ITS was lower than for 16S, the read utilization ratio ranged between 84% and 99% (Appendix A). Paired end reads were spliced into tags through the overlapping relationship between reads. A total of 5,617,159 tags were obtained for all 16S samples, with an average of 62,412 tags per sample, with an SD value of 2816. The average tag length was 268 bp, and the SD value was 23 bp (Appendix A). For ITS, there was a total of 5,438,323 tags for all samples combined after removing the primers. On average, each sample was 60,425 tags, with an SD value of 2930, and an average length of 219 bp, with an SD value of 22 (Appendix A). The clean tags were subsequently clustered into different operational taxonomic units (OTUs) at 97% similarity. A preliminary assessment of the abundance of OTUs suggested the species richness of the samples. A total of 5631 OTUs were identified from the 16S data, and 2236 OTUs were identified from the ITS data (Appendix A). The number of observed bacterial OTUs was much greater for the soil samples than fungal OTUs (Appendix A).

### 2.3. Analysis of the Microbial Community Composition between Treated and Untreated Soils

A comparative analysis of the microbial community composition between different sample groups was made to investigate whether there was an association between the treatments and changes in the microbiome. Principal component analysis (PCA) was used to construct 2D graphs to summarize the factors mainly responsible for differences in the OTU composition. The OTUs of bacterial community compositions clustered together (Figure 2A). Bacteria of D1C and D1T were different from D2C, D2T, D3C, and D3T, while there were no measurable differences in their community compositions between the four subsequent samples (Figure 2A). Bacterial communities in D2T and D3T were more closely clustered than in the control treatments, D2C and D3C (Figure 2A). However, obvious distinctions in community compositions were found among D1C, D1T, D2C, D2T, D3C, and D3T for fungi (Figure 2B). The differences in the community composition of fungi in D1T with D1C was smaller than variations observed between D2T and D3T, D2C and D3C (Figure 2B).

### 2.4. Microorganism Community Diversity in Soil Samples under Different Conditions

The identified OTUs were distributed across 35 bacterial phyla (Figure 3A). The predominant phyla were *Proteobacteria* and *Actinobacteria*, which together accounted for about 66% of the population, followed by *Acidobacteria* with 10.15% of OTUs (Figure 3A). Firmicutes, *Chloroflexi*, *WPS-2*, *Bacteroidetes*, *Gemmatimonadetes,* and *TM7* were recovered with a small proportion at the phylum level (Figure 3A). *Proteobacteria* was enriched highest in D3T among all six groups, especially compared with D3C (Figure 3A). The level of *Acidobacteria* was significantly higher in DT compared with DC, while the level of *Actinobacteria* was significantly less in DT than DC (Figure 3A). Furthermore, the bacteria detected corresponded to 31 different genera, without no significant enrichment, including *Acinetobacter*, *Bacillus*, *Leptothrix*, *Cupriavidus*, *Rhodoplanes*, and others (Figure 3B). The enrichment of several bacteria was reduced in DT compared with DC, and these included *Cryocola*, *Geodermatophilus*, and *Alicyclobacillus* (Figure 3B).

For the fungal kingdom, the OTUs spanned 13 phyla and 65 genera (Figure 4). From these, *Ascomycota* and *Basidiomycota* comprised 90% (Figure 4A). Interestingly, *Ascomycota* and *Basidiomycota* were both higher in natural soil (D1C and D1T) but reduced in D2T and D3T compared with D2C and D3C, respectively (Figure 4A). *Alternaria*, *Fusarium*, and *Humicola* formed the major proportion at the genus level (Figure 4B). The application of *B. subtilis* strain Bv17 suppressed *Alternaria* and *Humicola* (Figure 4B). These results indicated that the microbial distribution differed between soils treated with *B. subtilis* strain Bv17 and untreated soils.

### 2.5. Similarities and Differences of OTUs between Six Groups of Soil Samples

The number of common and unique OTUs of multiple samples is shown in a Venn diagram. For bacteria, there were 2840 common OTUs among D1C, D2C, and D3C, and 3313 OTUs in D1T, D2T, and D3T. There were 2476 common OTUs between DC and DT, and 364 and 837 of these OTUs were unique to DC and DT, respectively (Figure 5A and Appendix A). A total of 4920 OTUs were discovered in DC (D1C, D2C, and D3C) and 5191 OTUs were discovered in DT (D1T, D2T, and D3T). Among the 4480 common to DC and DT, 440 were unique in DC and 711 were unique in DT (Figure 5B and Appendix A). These 711 specific OTUs in DT corresponded to differential abundance of related bacteria (Appendix A). For fungi, there were 935, 892, and 760 common OTUs between D1C and D1T, D2C and D2T, and D3C and D3T, respectively (Appendix A). There were 156 unique OTUs in DC, 224 unique OTUs in DT, and 559 common OTUs between DC and DT (Figure 6A). Finally, a total of 342 unique OTUs were found in DC (D1C, D2C, and D3C) and 503 unique OTUs were found in DT (D1T, D2T, and D3T), with 1391 common OTUs between DC and DT (Figure 6B). The 503 unique OTUs in DT corresponded to the differential abundance of related fungi (Appendix A).

Ace, Chao, Shannon, and Simpson indices were used to analyze the richness and diversity of the soil microbial community. The Chao and Ace in DT were higher than in DC both for bacteria and fungi (Appendix A). The value of Shannon and Simpson were small; Shannon was higher in DT than in DC, but Simpson was lower in DT than in DC without significance (Appendix A). Particularly, the Chao index and Shannon index of the soil samples were selected to show the detailed distribution in each group. The results showed that both the Chao and Shannon index were relatively stable among D1T, D2T, and D3T, while the Chao and Shannon were decreased from D1C to D3C similarly in both bacterial and fungal communities (Figure 5C,D and Figure 6C,D), which suggested a higher diversity of the bacterial and fungal community in the soil treated by *B. subtilis* strain Bv17.

### 2.6. Analysis of Bacteria Diversity of Potato by 16S Sequencing

The number of raw DNA sequence data and clean reads data for each potato sample of bacteria was about 70,000 reads, and more than 99% of the reads from each sample remained after filtering for quality and size (Appendix A). The number of OTUs was annotated at 97% similarity to the cluster. A total of 257 OTUs were produced among 16S from potato samples (Appendix A). The average coverage was more than 99%, and the index of alpha diversity, including Chao, Ace, Shannon, and Simpson from PT was not significantly different from PC (Appendix A). The microorganism community similarity in different samples of potato was determined by PCA. The bacterial community in PC appeared more similar, but the three samples of PT clustered separately from each other (Figure 7A). The bacterial OTUs were assigned into 17 phyla and five genera, after combining the species with abundance less than 0.5% into others. The dominant bacterial phylum across all samples was Cyanobacteria, accounting for >96% of the OTUs. The second dominant bacterial phylum across all samples was Proteobacteria, with nearly 3.5% in PC and 2.6% in PT (Figure 7B). The assigned five genera were *Agrobacterium*, *Leptothrix*, *Ochrobactrum*, *Ralstonia*, and *Sediminibacterium*, comprising about 40% (Figure 7C). Furthermore, the OTUs were integrated among PC (PC1 and PC2) and PT (PT1 and PT2). A total of 143 OTUs were discovered in PC, and 244 OTUs were found in PT. From these OTUs, 130 were matched both in PC and PT, 13 were unique in PC, and 114 were unique in PT (Figure 7D). The unique OTUs in PT were significantly higher than in PC, and the unique OTUs in PT were annotated into bacteria species (Appendix A).

### 2.7. Analysis of Potato Quality Measures

*Bacillus subtilis* strain Bv17 treated plants had a higher tuber weight compared with the untreated control (Figure 8A). The internal quality of potato tubers is often estimated by the starch, sugar, and protein content [30]. For the processing industry and cooking purposes, the accepted dry matter content is between 18% and 20% [30]. The dry matter content in PC was 18.4% lower than in PT (20.6%) (Figure 8B). In addition, the content of starch, protein, and reducing sugars was also higher in PT compared to tubers in PC (Figure 8C–E). Potato tubers are an important source of vitamin C, which serves as an antioxidant with health-promoting effects [30]. The vitamin C levels in PC and PT groups were 48.4 mg/100 g and 48.5 mg/100 g, respectively, and these were not statistically different (Figure 8F).

## 3. Discussion

Several studies have demonstrated that the application of biocontrol agents could influence the microbial community. For example, the application of *Pseudomonas fluorescens pc78* to the soil affected the microbial community structure in the tomato rhizosphere [31]. Similarly, the endophytic microbial abundance was significantly higher in the stem of maize grown in *Trichoderma asperellum* granules-treated soil, with reductions in deoxynivalenol (DON) and fumonisin B1 (FB1) accumulation [32]. The application of biocontrol can influence seasonal epiphytic microbial dynamics on grapevine leaves [33]. In our research, we found that the abundance of microbial community of soil changed after *Bacillus subtilis* strain Bv17 treatment. Soil samples from DC and DT showed no statistically significant differences in the number of bacteria and fungi (Figure 1B,C). Interestingly, the OTU numbers of both bacteria and fungi from D1C to D3C progressively decreased, but the number of OTU from D1T to D3T registered an increase (Appendix A, and Appendix A). In addition, the Chao and Ace indices for DT were higher than in DC both for bacteria and fungi (Appendix A). The values of Shannon and Simpson indices were small, with Shannon being higher in DT than in DC. In contrast, Simpson was lower in DT than in DC, and the differences were not significant (Appendix A). These results indicated that the bacterial and fungal richness (Ace and Chao) were markedly increased after the *Bacillus subtilis* strain Bv17 treatment; however, the differences were not statistically significant based on the Shannon and Simpson indices.

The microbiome outside of plant tissues performs various plant beneficial activities such as suppression of potential phytopathogens and promotion of plant growth. The microbiome of plants plays a crucial role in both plant and ecosystem health [34]. In the soil, high microbial activity reflects the intensity and direction of various biochemical activity and the direction of various biochemical reactions. Even a slight decrease in soil microbial diversity or a change in the structure and function could affect the availability and absorption of nutrients [35]. The rhizosphere microbiome is important for plant growth and health [36,37]. Plants can recruit protective microorganisms to enhance microbial activity to suppress pathogens in the rhizosphere upon pathogen or insect attack [36,37]. The research showed that the continuous cropping of maize seed production could increase pathogenic pathogens, making maize in danger of pathogen invasion [38]. Here, the changes in the soil microbiol community following the application of *B. subtilis* strain Bv17 affected potato tuber growth (Figure 8A). Whether the impacts on potato plant and tuber health were influenced by decreasing the harmful effects of pathogens or if the microbial enhancements of the rhizosphere environment brought about the changes in potato is unclear and teasing out these differences is currently underway.

The biocontrol of soilborne plant diseases in cultivated crops has been explored more intensely in recent years. *P. dispersa* strains were able to inhibit black rot in sweet potato plants [39]. The use of a combination of biocontrol strains as a potential strategy could limit the soft rot and blackening diseases caused by *D. dianthicola* on potato plants and tubers [40]. *Streptomyces violaceusniger* AC12AB could promote potato growth by decreasing potato common scab and increasing potato yield [41]. The effect of *Pseudomonas fluorescens*, *Bacillus subtilis*, *P. aeruginosa,* and *Trichoderma spp.* to enrich the growth and yield of potato crop and induce resistance against wilt disease caused by *Ralstonia solanacearum* [42]. The application of a beneficial microbial combination efficiently enhanced plant and soil health under biotic stress through improving the microbial community structure [43]. *Lactobacillus plantarum SLG17* and *Bacillus amyloliquefaciens FLN13* function as biocontrol agents on durum wheat at a stage from heading until anthesis against *Fusarium spp*. [44]. In our research, we found that the yield and quality of potato were correspondingly improved by *Bacillus subtilis* strain Bv17 treatment. The important quality traits of the potato tubers such as the starch, sugar, protein, and vitamin C contents improved in biocontrol treated plots (Figure 8), even though the bacterial community on potato was unaltered in treated and untreated plots.

## 4. Materials and Methods

### 4.1. Preparation and Dosage of Bacillus subtilis Strain Bv17 in Field

*Bacillus subtilis* strain Bv17 was cultured in LB medium at 37 °C for 48 h on a rotating shaker at 200 rpm. *Bacillus subtilis* strain Bv17 fermentation liquid was added to 1 kg of carrier diatomite until the diatomite was saturated. This mixture was air-dried and pulverized to obtain a finely powdered product with an adsorption capacity of 1.2 L/kg. Prior to the application of the product in the field, a wetting agent (sodium dodecyl sulfonate, sodium lignosulfonate) and a dispersant (sodium carboxymethyl cellulose) were added to obtain a wettable powder. Two kg wettable powder was evenly spread on 666.67 m^2^ and tilled in to serve as the biocontrol treatment. An area of equal size was treated with water, and it served as the untreated control.

### 4.2. Soil Treatment and Collection of Soil and Potato Sample

Potato was planted in Shandong province (37°45′ north latitude, 116°29′ east longitude, altitude 27 m), which is in the east of China. The average rainfall was 466 mm, and the average temperature was 14.5 °C in 2019. The experiment included two main treatments. The first included the application of *Bacillus subtilis* strain Bv17 to the soil prior to planting, and the second was treated with water to serve as untreated control. Potato seedings were planted on 24 February 2019. The row spacing of potatoes was 40.0 cm, and the plant spacing was 37.5 cm. The plot area was 72.0 m^2^ (9.0 m × 8.0 m). The samples of soil were named D1C (Control) and D1T (Treat). Samples were collected approximately every two months from each treatment (14 April 2019, and 8 June 2019), and these were named D2C/D2T and D3C/D3T, respectively. Fifteen samples were collected from the treated and untreated plots at each sampling for a total of 90 samples during the cropping cycle. Twenty potatoes were collected at crop maturity (8 June 2019) from treated (PT) and untreated control (PC) plots. Collected potatoes were used for 16S sequencing and determining quality parameters as described below.

### 4.3. DNA Extraction and PCR Analysis

The DNAsecure Plant Kit (Tiangen, Beijing, China) was used to extract the genomic DNA from the soil samples according to the manufacturer’s instructions. The final DNA elution was performed using sterile deionized water. DNA quality and quantity were measured by NanoDrop1000 (Thermo Fisher Scientific, Inc., Newark, DE, USA) and by agarose gel electrophoresis. Extracted DNA was stored at −80 °C. The V3V4 region of the 16S rRNA gene was amplified using the primer V3V4-F: ACTCCTACGGGAGGCAGCA and V3V4-R: GGACTACHVGGGTWTCTAAT. For fungal community analysis, the ITS sequence was amplified with primer ITS1-F: GGAAGTAAAAGTCGTAACAAGG and ITS1-R: GCTGCGTTCTTCATCGATGC.

### 4.4. Sequencing Analysis

Paired-end reads were generated with the Illumina HiSeq2500 platform. The raw reads were preprocessed by removing the adapter, ambiguous bases, and those with low complexity to obtain the clean reads [45]. For a pooling library with barcode samples mixed, the clean reads were assigned to corresponding samples by allowing 0 base mismatch to barcode sequences with in-house scripts. Then, the consensus sequences were generated by FLASH (Fast Length Adjustment of Short reads, v1.2.11) [46]. The tags were clustered to the Operational Taxonomic Units (OTUs) by using scripts from software USEARCH (v7.0.1090) [47]. OTUs were filtered by sequences that were unassigned and not assigned to the target species. The filtered OTUs were used for downstream analyses. Alpha diversity was calculated to determine the complexity of species diversity for individual samples through several indices, including observed species, Chao, Ace, Shannon, and Simpson [48]. The indices were calculated by Mothur (v1.31.2). The formulae for the calculation of each index can obtained at http://www.mothur.org/wiki/Calculators (accessed on 14 September 2021).

### 4.5. Detection of Quality of Potato Tubers

From the *Bacillus subtilis* strain Bv17 treated, and water treated plots, 18 representative potato tubers were collected to calculate the weight of each tuber [49]. Samples of potato tubers were drying at 105 °C for 30 min and then at 80 °C to constant weight [49]. Furthermore, the starch content, protein content, reducing sugar content, and vitamin C content were determined following the procedures in the literature [30].

### 4.6. Statistical Analysis

Analysis of variance (ANOVA) was run on the sequencing data after testing the data for normality. Treatments were compared using Tukey’s test. Differences were considered significant at *p* < 0.05. In addition, the statistical analysis of the Chao and Shannon index and quality of potato were compared between treatments by pairwise *t*-tests using SPSS (v.20.2).

## 5. Conclusion

In conclusion, the diversity of soil microbial community compositions was significantly different in treated soil relative to the untreated control. While the bacterial community diversity on the skin of potato was not altered, tuber quality improved, nonetheless. The results indicated that the variation of soil microbial community by *Bacillus subtilis* strain Bv17 treatment was able to decrease diseases of potato and improve the quality and quantity of yield, and thus, it may shed light on the regulatory roles of *Bacillus subtilis* strain Bv17 in the production of crops.

## Figures and Tables

**Figure 1 ijms-22-12065-f001:**
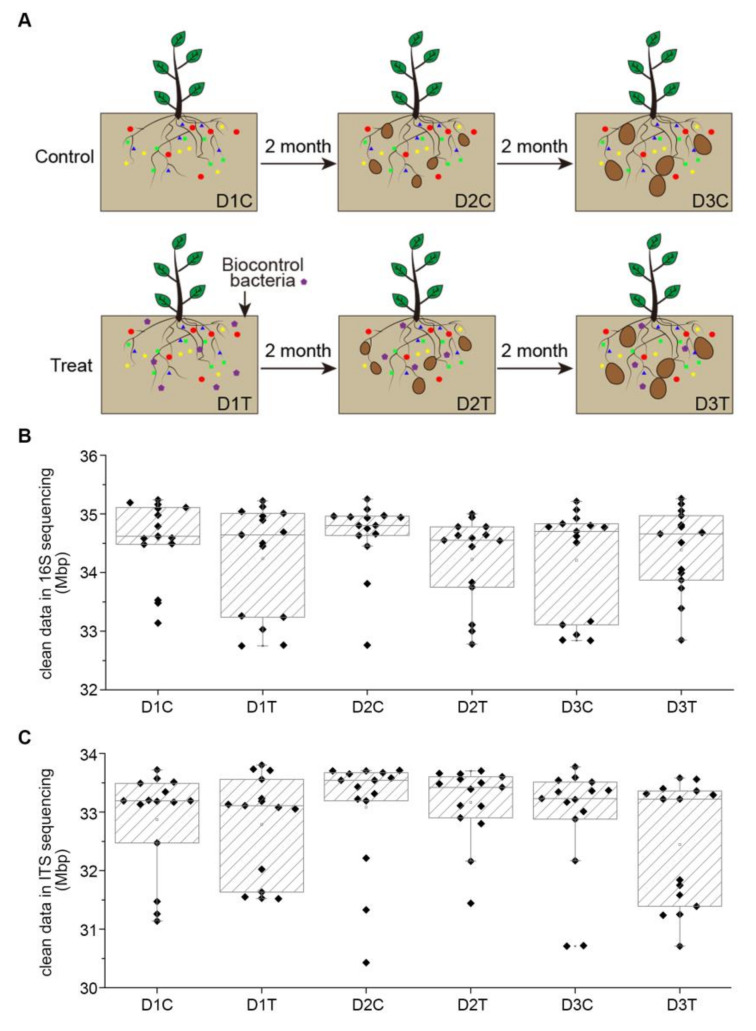
Characteristic of biocontrol treatment experiment and high-through sequencing. (**A**) Schematic diagram of experimental design. (**B**) The depth of 16S sequencing for each group of soil samples. (**C**) The depth of ITS sequencing for each group of soil samples.

**Figure 2 ijms-22-12065-f002:**
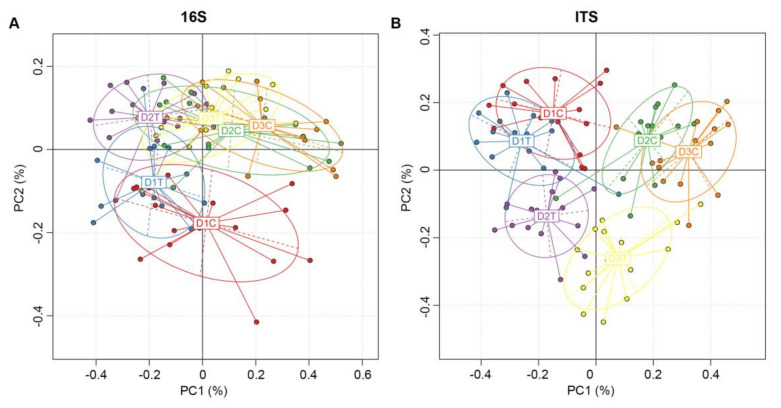
Principal component analysis based on OTU abundance in bacteria (**A**) and fungi (**B**). X-axis represents the mean principal component 1 and Y-axis represents the mean principal component 2. Numbers within parentheses represents contributions of principal components to differences among samples. Each dot represents a soil sample, and different colors represent different groups.

**Figure 3 ijms-22-12065-f003:**
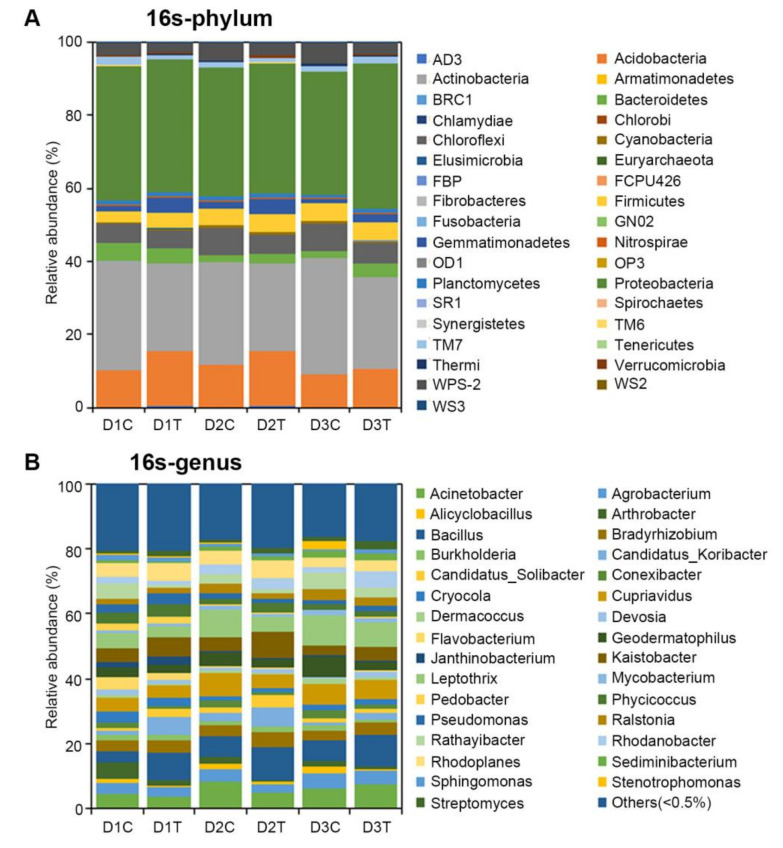
Relative abundances (percentage of sequences) of bacteria at phylum (**A**) and genus (**B**) levels in each group.

**Figure 4 ijms-22-12065-f004:**
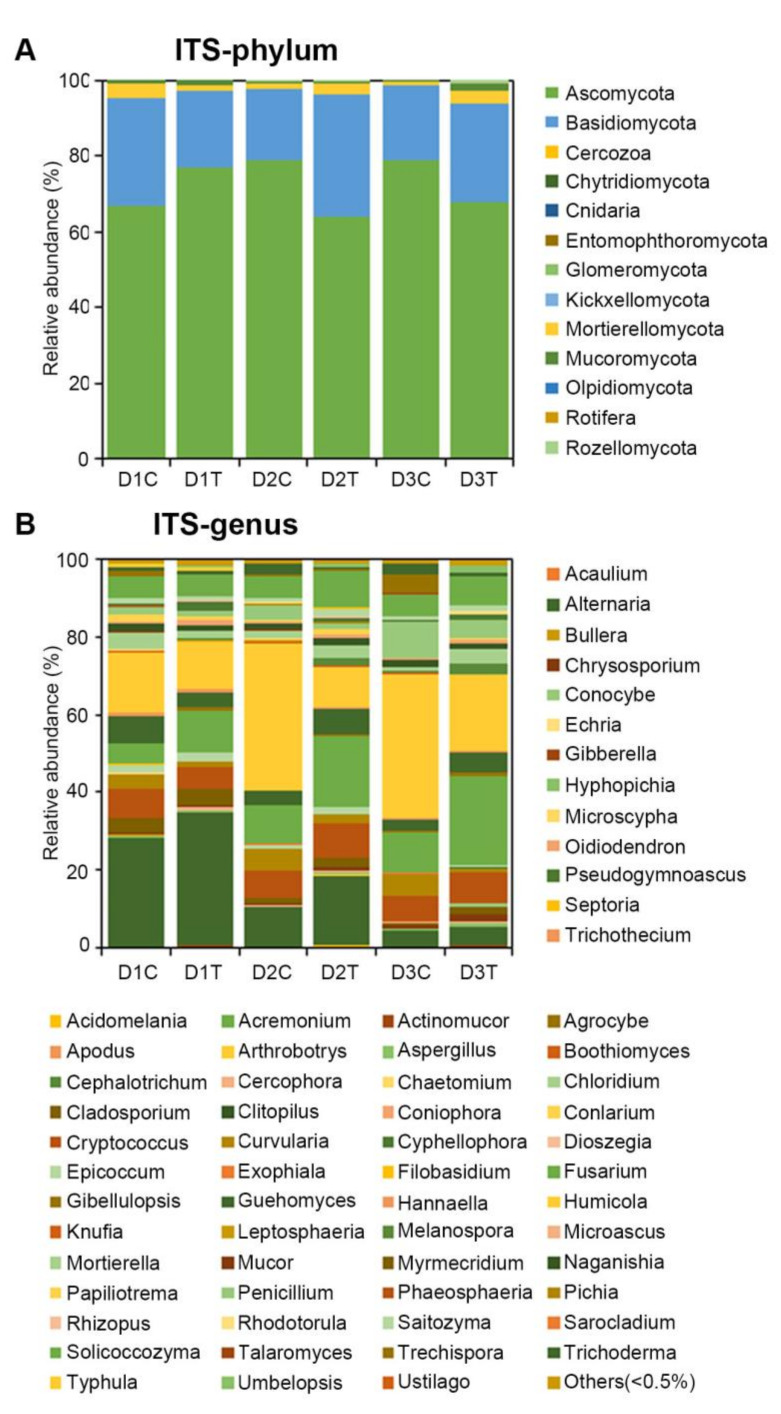
Relative abundances (percentage of sequences) of fungi at phylum (**A**) and genus (**B**) levels in each group.

**Figure 5 ijms-22-12065-f005:**
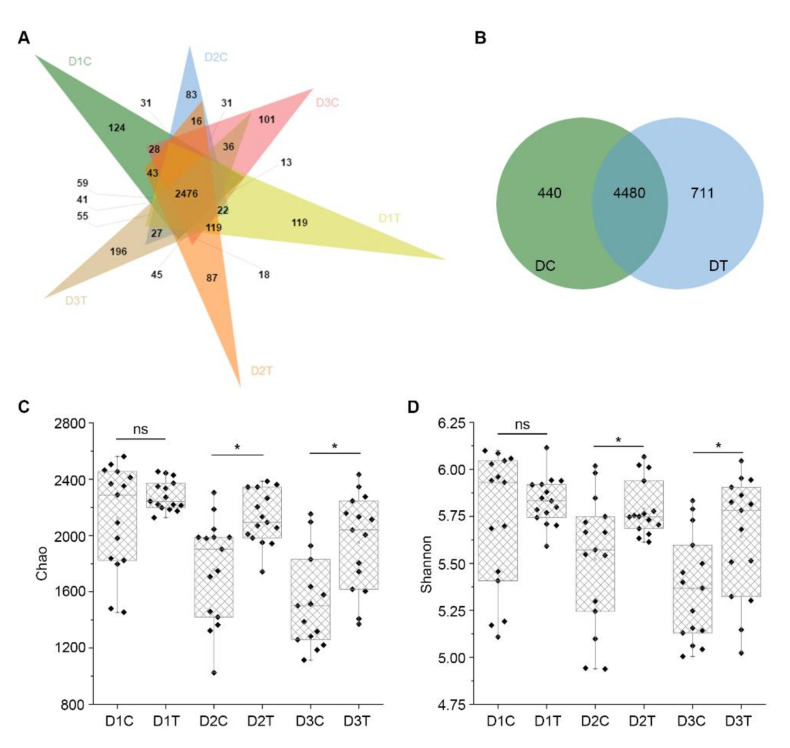
Analysis of bacterial diversity in soil samples. (**A**) Venn diagram of OTUs between each soil group in bacteria. DT refers to soil samples derived from plots treated with *Bacillus subtilis* strain Bv17, and those that came from plots treated with water are labeled DC. Soil samples were collected at two-month intervals three times (D1C, D1T, D2C, D2T, D3C, and D3T). (**B**) Venn diagram of OTUs between DC and DT. (**C**) The box diagram of alpha diversity of the Chao index in each soil groups. (**D**) The box diagram of alpha diversity of the Shannon index in each soil groups. Asterisks indicate statistically significant differences at *p* < 0.05 according to pairwise *t*-tests, and ns means no significant difference between treatments.

**Figure 6 ijms-22-12065-f006:**
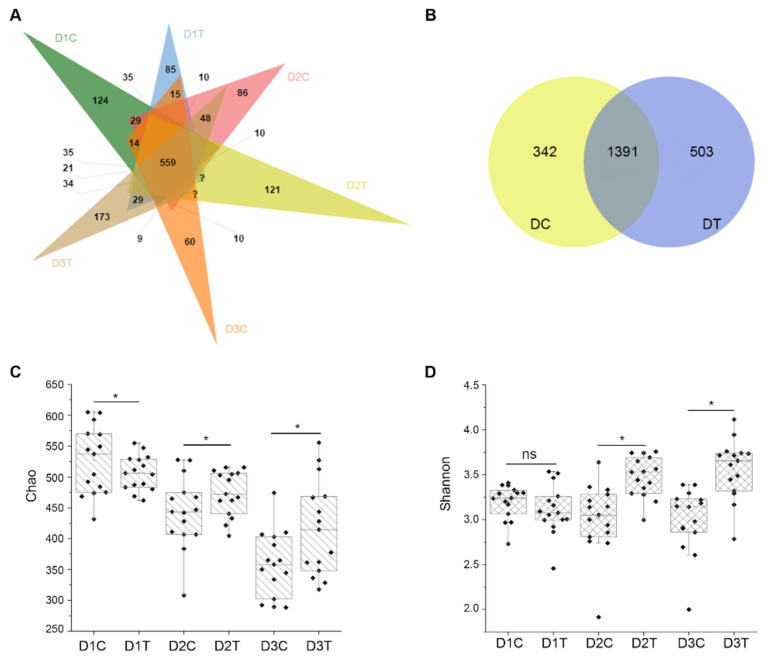
Analysis of fungal diversity in soil samples. (**A**) Venn diagram of OTUs between each soil groups in fungi. DT refers to soil samples derived from plots treated with *Bacillus subtilis* strain Bv17, and those that came from plots treated with water are labeled DC. Soil samples were collected at two-month intervals three times (D1C, D1T, D2C, D2T, D3C, and D3T). (**B**) Venn diagram of OTUs between DC and DT. (**C**) The box diagram of alpha diversity of Chao index in each soil group. (**D**) The box diagram of alpha diversity of the Shannon index in each soil groups. Asterisks indicate statistically significant differences at *p* < 0.05 according to pairwise *t*-tests, and ns means no significant differences between treatments.

**Figure 7 ijms-22-12065-f007:**
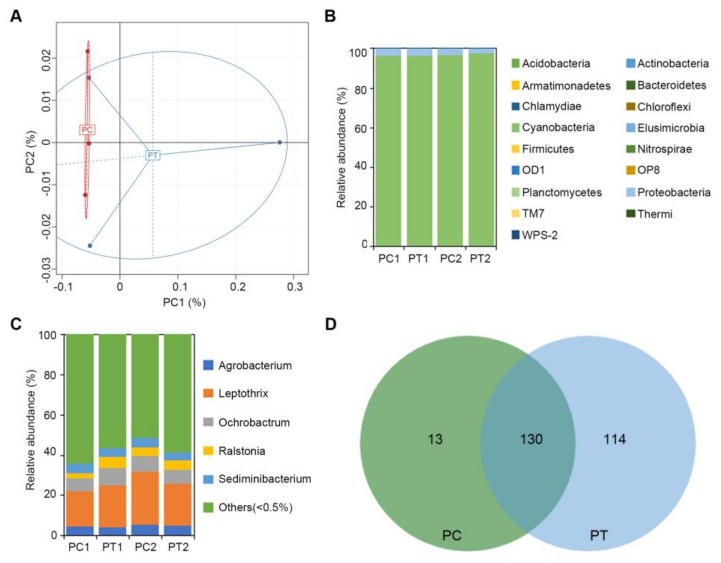
Characteristics of 16S sequencing data of potato samples. (**A**) Principal component analysis based on OTU abundance in bacteria. X-axis represents principal component 1 and Y-axis represents principal component 2. Numbers within parentheses represent contributions of principal components to differences among samples. Each dot represents an individual soil sample, and different colors represent different groups. (**B**) Relative abundances (percentage of sequences) of bacteria at the phylum level. (**C**) Relative abundances (percentage of sequences) of bacteria at the genus level. (**D**) Venn diagram of OTUs between PC and PT. PC refers to potatoes derived from the control treatment, and PT refers to potatoes derived from plots treated with *Bacillus subtilis* strain Bv17.

**Figure 8 ijms-22-12065-f008:**
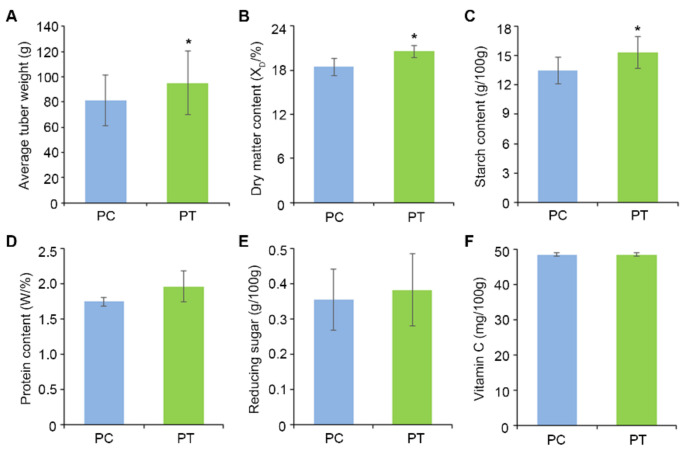
Effect of biocontrol treatment on potato tuber yield and quality. (**A**) Bar chart of average tuber weight. *n* = 18 (**B**) Bar chart of dry matter. (**C**) Bar chart of starch content. (**D**) Bar chart of protein content. (**E**) Bar chart of reducing sugar content. (**F**) Bar chart of vitamin C content. Asterisks indicate statistically significant differences at *p* < 0.05 according to *t*-test.

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
