# Peer review of "Rhizosphere Microbiomes of Potato Cultivated under Bacillus subtilis Treatment Influence the Quality of Potato Tubers"

_ijms, 2021, doi:10.3390/ijms222112065_

Round 1

Reviewer 1 Report

Section Introduction line 101 It would be good to give the aim of the research, a hypothesis

Lines 101-107 This part of the text can be treated as a summary. It should be at the end of the discussion.

Lines 119-131  section 2.1. Experimental design and treatment structure should be placed in the section 4. Materials and Methods

Figure 2  - The figure description is insufficient. An explanation of the abbreviations in the figure should be provided

This also applies to the other figures

Author Response

Comment 1

Section Introduction line 101 It would be good to give the aim of the research, a hypothesis

Answer:

We have added the objective for the study as “To identify the influences of biocontrol treatment with Bacillus subtilis strain Bv17 on microbial communities in soil and discover the relationship between microbial communities and quality of potatoes”. In addition, we have rewritten the introduction according to reviewer’s suggestion.

Comment 2

Lines 101-107 This part of the text can be treated as a summary. It should be at the end of the discussion.

Answer:

Thank you for your helpful suggestion. We have changed this part in the revised manuscript.

Comment 3

Lines 119-131 section 2.1. Experimental design and treatment structure should be placed in the section 4. Materials and Methods

Answer:

We exhibited the schedule and experimental design and treatment structure individually as one result in section 2.1 aiming at easier understanding for readers. Your comment is valuable. However, we applied for exhibition as one result, combined with the data of ITS and 16S sequencing also shown in Figure 1. We would like to modify if you still consider it is unsuitable.

Comment 4

Figure 2- The figure description is insufficient. An explanation of the abbreviations in the figure should be provided

This also applies to the other figures

Answer:

We have revised according to your comments.

Reviewer 2 Report

IJMS-1402305

Rhizosphere Microbiomes of Potato Cultivated under Bacillus subtilis Treatment Influence the Quality of Potato Tubers

Song et al.

The overall structure of this paper is unusual, with the Methods placed at the end after a short Discussion; most of the length of this paper is in the Results section. Some aspects of the Results were unclear until I read the details of the Methods section, and I usually prefer to read papers in the order of Introduction – Methods – Results – Discussion specifically to avoid that kind of confusion.

The Introduction is very good, clear and well-written. It needs additional information, particularly regarding the biocontrol used in this study, Bacillus subtilis Bv17.

The Methods are inadequate. Insufficient detail is presented regarding nearly every aspect of this study. The study site is not well identified, and no mention is made of climate and weather that would have a strong influence on the microbial community in the soil as well as the potato crop. Statistical tests are very briefly mentioned, but do not appear to have been rigorously applied to the main results. Some of the data-handling procedures, such as what was done with rare OTUs, are highly unusual yet not explained.  

The Results section is very long, and quite often redundant with identical information listed in the text and appearing in Figures. Potentially important differences between treatment and control are merely mentioned, and only rarely are the significances (or otherwise) of these differences indicated. Four diversity indices are employed, but there is no description of these indices, or why and how they were used. Sometimes, these diversity indices showed significant differences between treatment and control, and sometimes not – what does this mean about the microbial communities studied?

The Discussion is effectively missing, with only two paragraphs, one of which is a simple restatement of the main results (without further analysis or elaboration) and with little context, and the other paragraph is a short Conclusion that relies on insufficiently-described patterns.

The supplemental files consist largely of almost-meaningless lists of samples and raw data that could, perhaps, be packaged into a data package for publication.

Detailed comments follow.

  1. Introduction

The Results section includes mention of suppression of some fungal genera by the applied biocontrol organism. Was B. subtilis strain Bv17 developed specifically to target these fungi, or related fungi? Why choose to apply this biocontrol in this study?

  1. Results

LN 110-111 – please cite your relevant previous studies in this sentence.

Section 2.1. Experimental design and treatment structure belongs in the Methods section, not Results.

LN 145 – how were clean reads assigned to OTUs? What degree of similarity was used as the criterion to include or exclude sequences?

LN 162-163 – what does "were nearly similar" mean? Similarity is quantitative, so there must be a number that counts as "similar" and another number that counts as "not similar".

Are the differences (or lack of differences) described in this paragraph statistically significant? Similarity (or dissimilarity) matrices can be robustly compared.

LN 168-169 – this sentence is unclear. What studies? What single study?

LN 175-176 – how does the word "obviously" used here relate to statistical conventions such as "significant"? What does "obviously" mean here?

LN 189-190 – was B. subtilis Bv17 developed specifically to suppress Alternaria and/or Humicola?

LN 197-199 – the Venn diagram is mentiond, then relegated to the Supplemental Information. If a figure is important enough to mention in the main text, it is important enough to be included in the main paper, not hidden in Supplemental.

2.5. Similarities and Differences of OTUs between six groups of soil samples.

This paragraph is difficult to read and understand because it largely consists of sample names (D1C, D2C etc) and numbers rather than text. Rather a dense list of such things, I would prefer the data here represented in a table. The numbers within Figure 5A and 5B (and 6A and 6B) are quite a bit clearer than those same numbers buried in this paragraph. Please simplify and reduce redundancy.

Furthermore, the description of diversity indices should be moved to a separate paragraph.

LN 253-254 – why were rare species (less than 0.5%) combined into other, presumably unrelated species, instead of deleted?

2.7. Analysis of potato quality measures

Were any of the differences here statistically significant? The error bars in Figure 8 mostly overlap, suggesting the differences were not significant. Only Fig 8F, vitamin C content, is described as non-signficant. What tests were used? What p-values (or other criteria) were used to determine significance?

A large part of this paper's claim rests on the report that treatment with the biocontrol increased crop yield and quality – but if those increases are small relative to the variance within and among treatments, then they are much less important.

  1. Discussion

There is very little actual discussion in the Discussion section. The first – of only 2 paragraphs! – would fit in the Results section, as it merely re-states patterns. Furthermore, what is the point of reporting four different diversity indices – Shannon, Simpson, Ace, Chao – if the different results from each is simply reported and not further analysed?

The second paragraph seems to be the Conclusions, but without stronger statistical support, the conclusions are not justified.

  1. Materials and Methods.

LN 316 – this sentence is unclear. Was the study site near a city (named?)? Do you have coordinates (latitude, longitude) for the study site? Elevation? Climate conditions? What were the weather conditions during this growing season? The introduction mentions the importance of fertilisation for potatoes – were these potatoes fertilised? Irrigated?

LN 338 – Illumina HiSeq and Illumina MiSeq are distinct procedures – which was used and how in this study?

LN 352 – OK, this answers my earlier question about statistical tests. But my earlier point about indicating which differences (e.g. between treatment and control for potato nutrient contents) were significant or not remains.

Author Response

Comment 1

The overall structure of this paper is unusual, with the Methods placed at the end after a short Discussion; most of the length of this paper is in the Results section. Some aspects of the Results were unclear until I read the details of the Methods section, and I usually prefer to read papers in the order of Introduction – Methods – Results – Discussion specifically to avoid that kind of confusion.

Answer:

We confirmed the manuscript order of the journal IJMS again. The order is Introduction – Results – Discussion – Methods. We will be happy to change the order if necessary.

Comment 2

The Introduction is very good, clear, and well-written. It needs additional information, particularly regarding the biocontrol used in this study, Bacillus subtilis Bv17.

Answer:

We added the information of Bacillus subtilis strain Bv17 in the part of introduction.

Comment 3

The Methods are inadequate. Insufficient detail is presented regarding nearly every aspect of this study. The study site is not well identified, and no mention is made of climate and weather that would have a strong influence on the microbial community in the soil as well as the potato crop. Statistical tests are very briefly mentioned, but do not appear to have been rigorously applied to the main results. Some of the data-handling procedures, such as what was done with rare OTUs, are highly unusual yet not explained.  

Answer:

Thank you for your grateful comment. We rewritten the methods in revised draft.

Comment 4

The Results section is very long, and quite often redundant with identical information listed in the text and appearing in Figures. Potentially important differences between treatment and control are merely mentioned, and only rarely are the significances (or otherwise) of these differences indicated. Four diversity indices are employed, but there is no description of these indices, or why and how they were used. Sometimes, these diversity indices showed significant differences between treatment and control, and sometimes not – what does this mean about the microbial communities studied?

Answer:

We have rewritten the part of results in resubmitted manuscript.

Comment 5

The Discussion is effectively missing, with only two paragraphs, one of which is a simple restatement of the main results (without further analysis or elaboration) and with little context, and the other paragraph is a short Conclusion that relies on insufficiently-described patterns.

Answer:

We rewritten the discussion in revised draft.

Comment 6

The supplemental files consist largely of almost-meaningless lists of samples and raw data that could, perhaps, be packaged into a data package for publication.

Answer:

We have changed them.

Comment 7

The Results section in Introduction includes mention of suppression of some fungal genera by the applied biocontrol organism. Was B. subtilis strain Bv17 developed specifically to target these fungi, or related fungi? Why choose to apply this biocontrol in this study?

Answer:

We apologize for the lack of clarity. The Bacillus subtilis strain Bv17 was isolated from cotton planted at a disease nursery of Xinjiang, which could efficiently control Verticillium wilt. Therefore, we chose Bacillus subtilis strain Bv17 as biocontrol to discover its function in cultivated potato. We added this information and rewritten this part in introduction according to your suggestion.

Comment 8

 LN 110-111 – please cite your relevant previous studies in this sentence.

 Answer:

We are sorry for this mistake. We cited the previous study in revised draft.

Comment 9

Section 2.1. Experimental design and treatment structure belongs in the Methods section, not Results.

Answer:

We exhibited the schedule and experimental design and treatment structure individually as one result in section 2.1 aiming at easier understanding for readers. Your comment is valuable. However, we applied for exhibition as one result, combined with the data of ITS and 16S sequencing also shown in Figure 1. We would like to modify if you still consider it is unsuitable.

Comment 10

LN 145 – how were clean reads assigned to OTUs? What degree of similarity was used as the criterion to include or exclude sequences?

 Answer:

We are sorry for unclear classification. It was the clean tags subsequently clustered into different operational taxonomic units (OTUs) at 97% similarity. We corrected this sentence in new manuscript.

Comment 11

LN 162-163 – what does "were nearly similar" mean? Similarity is quantitative, so there must be a number that counts as "similar" and another number that counts as "not similar". Are the differences (or lack of differences) described in this paragraph statistically significant? Similarity (or dissimilarity) matrices can be robustly compared.

 Answer:

We would like to show the difference between D1C and D1T, D2C and D2T, D3C and D3T. We have corrected this sentence as “The difference of community composition of fungi in D1T with D1C was smaller than variations observed between D2T and D3T, D2C and D3C (Figure 2B)”.

Comment 12

LN 168-169 – this sentence is unclear. What studies? What single study?

 Answer:

We deleted this sentence in revised version.

Comment 13

LN 175-176 – how does the word "obviously" used here relate to statistical conventions such as "significant"? What does "obviously" mean here?

 Answer:

We statistic these data finding that this difference between DC and DT was significant. Therefore, we changed this word in new manuscript.

Comment 14

LN 189-190 - was B. subtilis strain Bv17 developed specifically to suppress Alternaria and/or Humicola?

 Answer:

Thank you for your valuable comments. Bacillus subtilis strain Bv17 specifically suppressed Alternaria and Humicola. We corrected it in resubmitted draft.

Comment 15

LN 197-199 – the Venn diagram is mentioned, then relegated to the Supplemental Information. If a figure is important enough to mention in the main text, it is important enough to be included in the main paper, not hidden in Supplemental.

 Answer:

The Venn diagrams in supplementary were the additional information of the Venn diagram in main text (Figure 5B and Figure 6B). We have rewritten this part in revised draft according to your suggestion.

Comment 16

2.5. Similarities and Differences of OTUs between six groups of soil samples.

This paragraph is difficult to read and understand because it largely consists of sample names (D1C, D2C etc) and numbers rather than text. Rather a dense list of such things, I would prefer the data here represented in a table. The numbers within Figure 5A and 5B (and 6A and 6B) are quite a bit clearer than those same numbers buried in this paragraph. Please simplify and reduce redundancy.

Furthermore, the description of diversity indices should be moved to a separate paragraph.

 Answer:

We appreciate this comment, which has allowed us to better focus the manuscript. We have rewritten this part in revised draft according to your suggestion.

Comment 17

LN 253-254 – why were rare species (less than 0.5%) combined into other, presumably unrelated species, instead of deleted?

 Answer:

Thank you for this comment. Abundances of species less than 0.5% combined into “other” of bacteria occupied the majority proportion at the genus level. Several of them are related species, so we did not delete them.

Comment 18

2.7. Analysis of potato quality measures

Were any of the differences here statistically significant? The error bars in Figure 8 mostly overlap, suggesting the differences were not significant. Only Fig 8F, vitamin C content, is described as non-significant. What tests were used? What p-values (or other criteria) were used to determine significance?

A large part of this paper's claim rests on the report that treatment with the biocontrol increased crop yield and quality – but if those increases are small relative to the variance within and among treatments, then they are much less important.

 Answer:

We have added statistical analysis in resubmitted draft. The average weight of potato tuber improved from 81.34 g to 95.05 g and this was statistically significant. Therefore, the claim that biocontrol increased crop yield.

Comment 19

Discussion

There is very little actual discussion in the Discussion section. The first – of only 2 paragraphs! – would fit in the Results section, as it merely re-states patterns. Furthermore, what is the point of reporting four different diversity indices – Shannon, Simpson, Ace, Chao – if the different results from each is simply reported and not further analyzed?

Answer:

Thank you for this valuable and remarkable comment. We have rewritten the part of discussion according to your suggestion.

Comment 20

The second paragraph seems to be the Conclusions, but without stronger statistical support, the conclusions are not justified.

 Answer:

Thank you for this valuable and remarkable comment. We have rewritten the part of discussion according to your suggestion.

Comment 21

Materials and Methods.

LN 316 – this sentence is unclear. Was the study site near a city (named?)? Do you have coordinates (latitude, longitude) for the study site? Elevation? Climate conditions? What were the weather conditions during this growing season? The introduction mentions the importance of fertilisation for potatoes – were these potatoes fertilised? Irrigated?

 Answer:

We have rewritten this part of method according to your suggestion.

Comment 22

LN 338 – Illumina HiSeq and Illumina MiSeq are distinct procedures – which was used and how in this study?

 Answer:

We are sorry for this mistake. Illumina HiSeq was used to sequence bacteria and fungi community. We have corrected it in revised manuscript.

Comment 23

LN 352 – OK, this answers my earlier question about statistical tests. But my earlier point about indicating which differences (e.g. between treatment and control for potato nutrient contents) were significant or not remains.

Answer:

We reanalyzed the data on the quality of potato tubers, finding that the yield, dry matter, and starch were significantly different as stated in the conclusion. We have rewritten the part of methods in resubmitted manuscript.

Reviewer 3 Report

Comments to the Authors

GENERAL COMMENTS

The manuscript titled “Rhizosphere Microbiomes of Potato Cultivated under Bacillus subtilis Treatment Influence the Quality of Potato Tubers” is quite interesting. The study characterized the fungal and bacterial microbiome in soil after following Bacillus subtilis Bv17 during the potato growth cycle.

Unfortunately, in my opinion, there are chapters of the manuscript that need to be improved. The introduction to the manuscript does not provide much relevant information. Does not describe the latest data on the issue. However, I have many more reservations about a discussion that has not actually been undertaken. This is just a summary of the research results. The authors of the manuscript referred to only three studies and this is not a discussion of the results obtained. The research methods and results are described correctly.

SPECIFIC COMMENTS

Abstract

The introductory sentence does not emphasize the importance of research. It is not advisable to explain the ITS abbreviation. It is too obvious.

keywords

 "OTU" is not necessary

“Introduction” chapter 

line 70-83, page 2

This information is too obvious to be devoted to a paragraph in the introduction chapter.

line 84-94, page 2

This information is not very revealing. They decrease the value of the manuscript.

line 297, page 10

Is the unit notation correct?

“Discussion” chapter

It is only on these sentences that the discussion is built. That's all:

“Even a slight decrease in soil microbial diversity or a change in the structure and function could affect the availability and absorption of nutrients[27].” (lines 288-290, page 10)

“Rhizosphere microbiome is important for plant growth and health[28, 29]” (line 305, page 10).

Unfortunately, in my opinion, this is not a scientific discussion. This chapter particularly reduces the value of the manuscript.

Author Response

Comment 1

Abstract: The introductory sentence does not emphasize the importance of research. It is not advisable to explain the ITS abbreviation. It is too obvious.

Answer:

Thank you for your suggestion. We corrected this in revised draft.

Comment 2

Keywords: "OTU" is not necessary

Answer:

Thank you for your suggestion. We deleted this keyword in revised draft.

Comment 3

line 70-83, page 2

This information is too obvious to be devoted to a paragraph in the introduction chapter.

Answer:

We deleted this paragraph, and added the introduction of application and function of biocontrol in introduction chapter in resubmitted manuscript.

Comment 4

line 84-94, page 2

This information is not very revealing. They decrease the value of the manuscript.

Answer:

Thank you for your remarkable comment. We rewritten this part in introduction in resubmitted manuscript.

Comment 5

line 297, page 10

Is the unit notation correct?

Answer:

Thank you for this comment. We checked the unit notation. It is correct.

Comment 6

It is only on these sentences that the discussion is built. That's all:

“Even a slight decrease in soil microbial diversity or a change in the structure and function could affect the availability and absorption of nutrients[27].” (lines 288-290, page 10)

“Rhizosphere microbiome is important for plant growth and health[28, 29]” (line 305, page 10).

 Answer:

We have rewritten discussion in resubmitted manuscript.

Unfortunately, in my opinion, this is not a scientific discussion. This chapter particularly reduces the value of the manuscript.

Answer:

We have rewritten discussion in resubmitted manuscript.

Round 2

Reviewer 2 Report

IJMS-1402305-V2

Song et al.

I am impressed by the improvements made to this revised version of the manuscript. It is considerably more clear overall. The new and revised figures and figure captions are also much better. I particularly like the added paragraph in the introduction about biocontrol agents (LN 71-87).

Some parts of the Methods remain unclear. LN 351 – as written, this sentence mentions a city named "East" locaed in China. I think it would be more clear, if this is accurate, as "Potato was planted in Shandong province (....north latitude,... east longitude... altitude), located in the eastern part of China."

Overall, I believe this manuscript is much improved and will be of interest to many readers interested in protecting crop plants with biocontrols.

Author Response

Reviewer 2

I am impressed by the improvements made to this revised version of the manuscript. It is considerably more clear overall. The new and revised figures and figure captions are also much better. I particularly like the added paragraph in the introduction about biocontrol agents (LN 71-87).

Some parts of the Methods remain unclear. LN 351 – as written, this sentence mentions a city named "East" locaed in China. I think it would be more clear, if this is accurate, as "Potato was planted in Shandong province (....north latitude,... east longitude... altitude), located in the eastern part of China."

Overall, I believe this manuscript is much improved and will be of interest to many readers interested in protecting crop plants with biocontrols.

Answer:

Thank you for your valuable comment. We rewritten this part in modified manuscript.

Reviewer 3 Report

Comments to the Authors

The manuscript chapters are correctly written. Moreover, in my opinion, the use of the reviewers' comments by the authors contributed to the value of the manuscript being much more significant. The authors used a wide variety of methods and conducted a comprehensive evaluation of them. This is an asset of the manuscript. The results of the research were presented in an innovative way. I still stand by my opinion that the discussion chapter, which deals with very few references, is the weakest point of the manuscript, nevertheless I recommend this manuscript to publication in International Journal of Molecular Sciences.

Author Response

Reviewer 3

The manuscript chapters are correctly written. Moreover, in my opinion, the use of the reviewers' comments by the authors contributed to the value of the manuscript being much more significant. The authors used a wide variety of methods and conducted a comprehensive evaluation of them. This is an asset of the manuscript. The results of the research were presented in an innovative way. I still stand by my opinion that the discussion chapter, which deals with very few references, is the weakest point of the manuscript, nevertheless I recommend this manuscript to publication in International Journal of Molecular Sciences.

Answer:

Thank you for your helpful comment. We modified the part of discussion and added five more references as follows:

Gobbi, A.; Kyrkou, I.; Filippi, E.; et al. Seasonal epiphytic microbial dynamics on grapevine leaves under biocontrol and copper fungicide treatments. Sci Rep. 2020, 10, 681.

Berg, G.; Koberl, M; Rybakova, D.; et al. Plant microbial diversity is suggested as the key to future biocontrol and health trends. FEMS Microbiol Ecol. 2017, 93.

Zhao, Y.; Fu, W.; Hu, C.; et al. Variation of rhizosphere microbial community in continuous mono-maize seed production. Sci Rep. 2021, 11, 1544.

Gupta, R.; Singh, A.; Srivastava, M.; et al. Plant-microbe interactions endorse growth by uplifting microbial community structure of Bacopa monnieri rhizosphere under nematode stress. Microbiol Res. 2019, 218, 87-96.

Baffoni, L.; Gaggia, F.; Dalanaj, N.; et al. Microbial inoculants for the biocontrol of Fusarium spp. in durum wheat. BMC Microbiol. 2015, 15, 242.
